# LAGRANGIAN FLUID SIMULATION WITH CONTINUOUS CONVOLUTIONS

**Benjamin Ummenhofer**
Intel Labs

**Lukas Prantl & Nils Thuerey**
Technical University of Munich

**Vladlen Koltun**
Intel Labs

## ABSTRACT

We present an approach to Lagrangian fluid simulation with a new type of convolutional network. Our networks process sets of moving particles, which describe fluids in space and time. Unlike previous approaches, we do not build an explicit graph structure to connect the particles but use spatial convolutions as the main differentiable operation that relates particles to their neighbors. To this end we present a simple, novel, and effective extension of N-D convolutions to the continuous domain. We show that our network architecture can simulate different materials, generalizes to arbitrary collision geometries, and can be used for inverse problems. In addition, we demonstrate that our continuous convolutions outperform prior formulations in terms of accuracy and speed.

## 1 INTRODUCTION

Understanding physics can help reasoning about our environment and interacting with it. Neural networks have emerged as a particularly promising approach to capture the complexity of natural phenomena from data (Ling et al., 2016; Tompson et al., 2017; Morton et al., 2018). An important aspect of learning physics with neural networks is the choice of representation. Lagrangian representations based on particles are particularly popular and have supported recent results with rigid bodies, deformable solids, and fluids (Battaglia et al., 2016; Mrowca et al., 2018; Li et al., 2019). Many of these approaches use graph structures to define interactions; the existence of an edge determines in a binary fashion whether two particles interact.

However, a wide range of physical processes such as fluid mechanics are described by continuous partial differential equations rather than discrete graph structures. The continuous, volumetric, and tightly coupled nature of these processes causes inherent difficulties for graph-based approaches, such as a large number of edge connections that must be established, tracked, and disengaged as the particles move. In this work, instead of using graphs as the underlying representation, we adopt a continuous viewpoint. We propose to use convolutional networks (ConvNets) with *continuous convolutions* on particles for learning fluid mechanics. We treat fluids as spatially continuous functions sampled at a finite set of (continuously evolving) positions and process them with a novel continuous convolution layer. This matches the continuous nature of the problem more closely and simplifies the definition of neural networks by abstracting the underlying particle representation.

We extend the grid-based filter representation commonly used for discrete convolutions to the continuous domain by simple linear interpolation. Linear interpolation of the filters allows efficient lookup of spatially varying filter values at arbitrary positions while retaining the compactness and efficiency of the grid representation. In addition, we use a window function to define the support of the filters and a ball-to-cube mapping to support spherical receptive fields. We show that our convolutions, despite their simplicity, perform better than more sophisticated representations (Wang et al., 2018; Schenck & Fox, 2018).

With the presented continuous convolution layer, we develop an efficient ConvNet architecture for learning fluid mechanics. The network processes sets of particles. We use dynamic particles to represent the fluid and static particles to describe the boundary of the scene. Modeling the scene boundary with particles makes it easy to apply our network to new scenes and allows the network to learn collision handling in a unified framework. Our network generalizes to arbitrary obstacle configurations and can simulate a range of material behavior. To demonstrate the usefulness of

a learned – and hence differentiable – fluid simulator, we show that material properties can be estimated from observed simulation data. Experimental results indicate that the presented approach outperforms a state-of-the-art graph-based framework (Li et al., 2019).

## 2 RELATED WORK

Fluids encompass a range of materials that are important in everyday life and throughout science and engineering (Wilcox, 2006). A prominent set of methods, known as Smoothed Particle Hydrodynamics (SPH), employs a Lagrangian viewpoint to simulate these phenomena. SPH originated in particle-based models for astrophysics (Gingold & Monaghan, 1977) and has become extremely popular for simulating complex flows in many fields of science (Monaghan, 1988). Among others, it has been highly successful for modeling interface flows (Colagrossi & Landrini, 2003), complex multi-phase phenomena (Hu & Adams, 2006), and even magnetohydrodynamics (Price, 2012).

SPH and its variants have been widely used to model complex real-world phenomena for visual effects. Following the earlier development of physics-based fluid simulation for special effects by Foster & Metaxas (1996), the introduction of SPH (Müller et al., 2003) has led to a large class of powerful algorithms (Solenthaler & Pajarola, 2009; Bender & Koschier, 2015). These applications often involve complex geometries at large scales, and extensions such as FleX (Macklin et al., 2014) and pressure-aware rigid-body coupling (Gissler et al., 2018) broaden the framework to encompass many physical phenomena.

Lagrangian flows have also been considered in machine learning. The pioneering work of Ladický et al. (2015) demonstrated that flow representations can be learned with regression forests. CNNs were used by Tompson et al. (2017) to accelerate the expensive pressure correction step of grid-based solvers, while other works have focused on super-resolution (Xie et al., 2018), learning time evolution via the Koopman operator (Morton et al., 2018), and learning reduced representations (Wiewel et al., 2019; Kim et al., 2019). Differentiable SPH solvers were proposed to solve control tasks for robotic applications (Schenck & Fox, 2018). Generic physics simulations for Lagrangian rigid and deformable bodies were considered in a series of works that developed graph-based representations (Battaglia et al., 2016; Sanchez-Gonzalez et al., 2018). Such graph-based representations were recently applied directly to fluid simulation (Mrowca et al., 2018; Li et al., 2019). We share with these recent works the goal of modeling Lagrangian fluids with differentiable neural networks, but take a different tack: rather than using graph-based representations, we work with point clouds and continuous convolutions over the spatial domain.

From a technical perspective, our approach is also related to existing works that apply convolutions on point clouds. A number of methods transferred the convolution concept to point clouds in the context of semantic classification and segmentation of 3D objects (Hua et al., 2018; Atzmon et al., 2018; Hermosilla et al., 2018; Li et al., 2018; Su et al., 2018; Wu et al., 2019; Xu et al., 2018; Lei et al., 2019). Particularly notable in our context is the work of Wang et al. (2018), who used continuous convolutions to compute the scene flow between two point clouds, and the aforementioned work of Schenck & Fox (2018), who used convolutions to implement a differentiable version of position-based fluids (Macklin & Müller, 2013). Both works define continuous convolution operators that can support regression tasks and we compare to them directly in Section 6. Most closely related to our filter representation is the work of Fey et al. (2018). They use B-splines to define continuous filters and propose to use spherical coordinates to implement spherical receptive fields. Spherical coordinates are problematic due to singularities and require special treatment, which we avoid with a ball-to-cube mapping. Furthermore, the output of the operator of Fey et al. (2018) can be discontinuous. We show in our ablation study in Section 6 that applying a window function to guarantee a continuous output is advantageous for our task.

## 3 BASICS

Fluids have been studied for centuries, and the Navier-Stokes equations for incompressible fluids are well established (Batchelor, 1967):

$$\frac{\partial \mathbf{v}}{\partial t} + \mathbf{v} \cdot \nabla \mathbf{v} = -\frac{1}{\rho}\nabla p + \nu \nabla^2 \mathbf{v} + \boldsymbol{g}, \quad \text{s.t. } \nabla \cdot \mathbf{v} = 0. \tag{1}$$

A common approach to solve these partial differential equations is to approximate the fluid with a set of smooth particles (Monaghan, 1988). Each particle corresponds to a continuous blob of matter and carries the local properties of the fluid, such as velocity and density, which move with the flow. This is motivated by the fact that a function $A(\mathbf{x})$ can be represented by an integral interpolation

$$A(\mathbf{x}) = \int A(\mathbf{x}')\delta(\|\mathbf{x} - \mathbf{x}'\|_2)dV(\mathbf{x}'), \tag{2}$$

where $\delta(x)$ denotes the Dirac delta function. This equation can be discretized as

$$A(\mathbf{x}) \approx \sum_i V_i A_i W(\|\mathbf{x} - \mathbf{x}'\|_2, h), \tag{3}$$

where $V_i$ is the volume at the given point in space and

$$\lim_{h \to 0} W(|\mathbf{x} - \mathbf{x}'|, h) = \delta(|\mathbf{x} - \mathbf{x}'|). \tag{4}$$

Here $W(x, h)$ is a smooth kernel or convolution with radius $h$, usually in the form of a Gaussian distribution, but more complex functions can also be used. In practice, the kernel is finite and yields localized neighborhoods of particles that interact via interactions weighted by the kernel of its derivatives. In this way, the continuous description for fluids from Equation 1 can be discretized in a Lagrangian fashion and solved numerically. Typically, internal forces are calculated based on the local pressure, viscosity, and surface tension, which give an update for the position of each particle. Below, we adopt the position-based fluids (PBF) method (Macklin & Müller, 2013; Macklin et al., 2014), which likewise is based on SPH, but reformulates the updates as constraints on the positions.

## 4 CONTINUOUS CONVOLUTIONS

The discrete convolution operator as commonly used in ConvNets is defined as

$$(f * g)(\mathbf{x}) = \sum_{\boldsymbol{\tau} \in \Omega} f(\mathbf{x} + \boldsymbol{\tau})g(\boldsymbol{\tau}), \tag{5}$$

where $f$ and $g$ are the input and the filter function, $\mathbf{x}$ is the position, $\boldsymbol{\tau}$ is the shift vector, and $\Omega$ is the set of shift vectors that defines the support of the filter function. On regular data such as images, the positions $\mathbf{x}$ range over a regular grid and the shift vectors $\boldsymbol{\tau}$ are integer-valued, i.e. $\mathbf{x}, \boldsymbol{\tau} \in \mathbb{Z}^d$ for some dimensionality $d$. Analogously in the continuous domain, this convolution is defined as

$$(f * g)(\mathbf{x}) = \int_{\mathbb{R}^d} f(\mathbf{x} + \boldsymbol{\tau})g(\boldsymbol{\tau})d\boldsymbol{\tau}, \tag{6}$$

where $\mathbf{x}$ and $\boldsymbol{\tau}$ are real-valued vectors, i.e. $\mathbf{x}, \boldsymbol{\tau} \in \mathbb{R}^d$.

We now adapt this definition to unstructured point clouds. In this setting we have a finite number of points that sample the function $f$ but do not lie on a grid. For a point cloud with $i = 1, .., N$ points with values $f_i$ at positions $\mathbf{x}_i$, we define the convolution at position $\mathbf{x}$ as

$$(f * g)(\mathbf{x}) = \frac{1}{\psi(\mathbf{x})} \sum_{i \in \mathcal{N}(\mathbf{x}, R)} a(\mathbf{x}_i, \mathbf{x}) \, f_i \, g(\Lambda(\mathbf{x}_i - \mathbf{x})). \tag{7}$$

$\mathcal{N}(\mathbf{x}, R)$ is the set of points within a radius $R$ around $\mathbf{x}$. $a$ is a scalar function that can be used for density normalization specific to the points $\mathbf{x}_i$ and $\mathbf{x}$ as in Hermosilla et al. (2018). In the simplest case, $a$ can be constant: $a = 1$. In our case we want to ensure a smooth response of our convolution under varying particle neighborhoods, therefore we define $a$ as a window function:

$$a(\mathbf{x}_i, \mathbf{x}) = \begin{cases} \left(1 - \frac{\|\mathbf{x}_i - \mathbf{x}\|_2^2}{R^2}\right)^3 & \text{for } \|\mathbf{x}_i - \mathbf{x}\|_2 < R \\ 0 & \text{else.} \end{cases} \tag{8}$$

A similar function has been used by Müller et al. (2003) in the SPH framework. $\psi$ is another scalar function for normalization, which can be set in our implementation as either

$$\psi(\mathbf{x}) = 1 \quad \text{or} \quad \psi(\mathbf{x}) = \sum_{i \in \mathcal{N}(\mathbf{x}, R)} a(\mathbf{x}_i, \mathbf{x}). \tag{9}$$

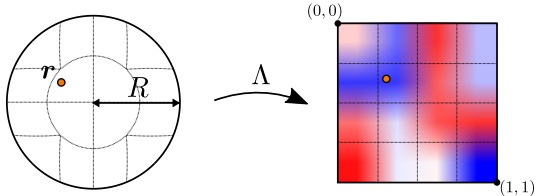

Figure 1: We use spherical filter shapes for our continuous convolutions but use regular grids to store the filter values. The left part of the figure shows a spherical region with radius $R$ and a point with relative coordinates $\mathbf{r}$ with respect to the center. We transform $\mathbf{r}$ via a mapping $\Lambda$ to normalized coordinates in a regular grid. The thin dotted lines illustrate the distortion of the mapping. To look up the final filter value we use trilinear interpolation in the regular grid.

We use $\psi(\mathbf{x}) = 1$, since changes in the density of particles are an important feature for simulating fluids.

For the filter function $g$ we simply use a regular grid to store the filter values but use linear interpolation to make $g$ a continuous function. In addition, we use a mapping $\Lambda(\mathbf{r})$ of a unit ball to a unit cube to implement spherical filters as shown in Figure 1. We use the mapping described by Griepentrog et al. (2008) and give the detailed function in the appendix. The intermediate coordinate mapping $\Lambda$ provides the flexibility to implement different spatial shapes while keeping the advantages of a regular grid for the storage and lookup of filter values.

Note that Equation 7 uses a similar approximation as in the SPH framework (Equation 3). Assuming that each point represents the same volume, $V_i$ is a constant factor in Equation 3, which we drop in our definition.

## 5 Learning Fluid Mechanics with Convolutional Networks

Our goal is to learn fluid mechanics from observing the motion of particles. The input to our ConvNet is a set of particles with corresponding features. Since position itself is not a feature but simply defines the particle's position in space, we must assign a feature vector to each particle. The feature vector we use is a constant scalar 1 accompanied by the velocity $\mathbf{v}$ and the viscosity $\nu$. A particle $p_i^n$ at timestep $n$ with its position and input feature vector is thus a tuple $(\mathbf{x}_i^n, [1, \mathbf{v}_i^n, \nu_i])$. Defining the velocity explicitly as an input feature allows us to compute intermediate velocities and positions as in Ladický et al. (2015) and to apply external forces and pass this information to the network. We compute the intermediate positions $\mathbf{x}_i^{n*}$ and velocities $\mathbf{v}_i^{n*}$ beginning with timestep $n$ with Heun's method as

$$\mathbf{v}_i^{n*} = \mathbf{v}_i^n + \Delta t \, \mathbf{a}_{\text{ext}} \tag{10}$$

$$\mathbf{x}_i^{n*} = \mathbf{x}_i^n + \Delta t \, \frac{\mathbf{v}_i^n + \mathbf{v}_i^{n*}}{2}. \tag{11}$$

The vector $\mathbf{a}_{\text{ext}}$ is an acceleration through which we can apply external forces to control the fluid or to simply apply gravity. The intermediate positions and velocities lack any interactions between particles or the scene, which we are going to implement with a ConvNet. To enable the network to handle collisions with the environment we define a second set of static particles $s_j$. We sample particles on the boundaries of the scene with normals $\mathbf{n}_j$ as the feature vectors, i.e. $s_j = (\mathbf{x}_j, [\mathbf{n}_j])$. Our network implements the function

$$[\Delta\mathbf{x}_1, \ldots, \Delta\mathbf{x}_N] = \text{ConvNet}(\{p_1^{n*}, \ldots, p_N^{n*}\}, \{s_1, \ldots, s_M\}), \tag{12}$$

which uses convolutions to combine features from both particle sets. $\Delta\mathbf{x}$ is a correction of the position which accounts for all particle interactions including the collision handling with the scene. Finally, we apply the correction to update positions and velocities for $n+1$ as

$$\mathbf{x}_i^{n+1} = \mathbf{x}_i^{n*} + \Delta\mathbf{x}_i \tag{13}$$

$$\mathbf{v}_i^{n+1} = \frac{\mathbf{x}_i^{n+1} - \mathbf{x}_i^n}{\Delta t}. \tag{14}$$

Note that the updated position $\mathbf{x}_i^{n+1}$ depends on the output vector $\Delta\mathbf{x}_i$ and allows us to directly define our learning objective on the particle positions.

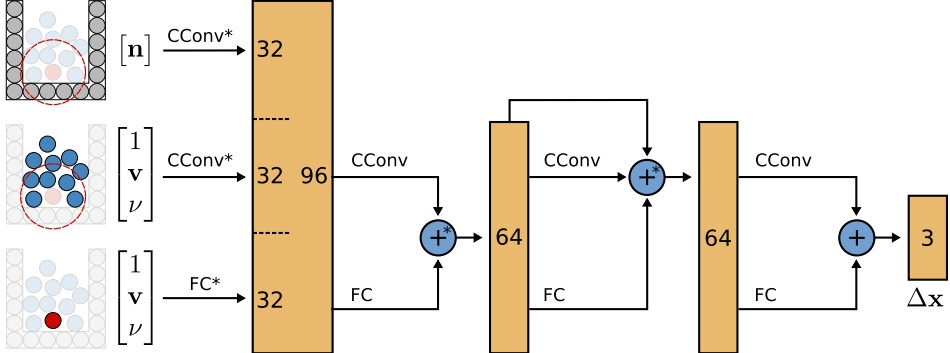

Figure 2: Schematic of our network with a depth of four. In the first depth level we compute convolutions at each dynamic particle location with the set of static particles that defines the environment as well as the dynamic particle set. We also directly process the features of each particle via a fully-connected stream. In the following levels, we compute convolutions only on the dynamic particles. At each level we use addition to aggregate the features computed by convolutions and fully-connected layers. Between the second and third level we also include a residual connection. The final level generates the position correction $\Delta \mathbf{x}$. Operations annotated with a * are followed by the ReLU activation function. All CConv and FC operations use an additive bias.

## 5.1 NETWORK ARCHITECTURE

We use a simple convolutional architecture with an effective depth of four. An overview of the network is shown in Figure 2. Since we want to compute the correction for all dynamic particles in our scene, we compute convolutions for the intermediate positions defined in Equation 11. Our network is a sequence of continuous convolutions (CConv), which are defined by an input particle set, the positions at which we want to evaluate the convolution, its filters $G$, and the radius $R$. For instance, to describe a convolution on the static particles $s_i$ at intermediate positions $\mathbf{x}_i^{n*}$ we can write

$$[\mathbf{f}_1, \ldots, \mathbf{f}_N] = \text{CConv}(\{s_1, \ldots, s_M\}, [\mathbf{x}_1^{n*}, \ldots, \mathbf{x}_N^{n*}], G, R), \tag{15}$$

where $\mathbf{f}_i$ are the computed output features for each position $\mathbf{x}_i^{n*}$. $G$ is a 5D array storing all filters in the layout $[\text{width}, \text{height}, \text{depth}, \text{ch}_{\text{in}}, \text{ch}_{\text{out}}]$. In contrast to discrete convolutions on a regular grid, the spatial filter dimensions here do not define the receptive field but the resolution of the filters. The receptive field depends only on the radius $R$, which specifies the spatial extent. Throughout our network we use filters with a spatial resolution of $[4, 4, 4]$ and a radius of $4.5$ times the particle radius.

For convolutions within the dynamic particles we exclude the particle at which we evaluate the convolution and instead process the particle's own features in a stream of fully-connected layers. After each depth level we then combine the result from the convolutions and the fully-connected layers by addition. This can be interpreted as a convolution with a spatial resolution of $4 \times 4 \times 4 + 1$. We found that this design improves accuracy and allows us to use smaller filters with even sizes (see Table 2).

## 5.2 TRAINING PROCEDURE

We train our fluid simulation network in supervised fashion based on particle trajectories produced by classic ("ground-truth") physics simulation. Our loss is defined as follows:

$$\mathcal{L}^{n+1} = \sum_{i=1}^{N} \phi_i \left\| \mathbf{x}_i^{n+1} - \hat{\mathbf{x}}_i^{n+1} \right\|_2^{\gamma}. \tag{16}$$

The ground-truth position at timestep $n + 1$ is denoted by $\hat{\mathbf{x}}_i^{n+1}$ and the predicted position from the network is denoted by $\mathbf{x}_i^{n+1} = \mathbf{x}_i^{n*} + \Delta \mathbf{x}_i$, where $\Delta \mathbf{x}_i$ is provided by the network. $\phi_i$ is an individual weight for each point. We use $\phi_i = \exp(-\frac{1}{c}|\mathcal{N}(\mathbf{x}_i^{n*})|)$, which emphasizes the loss for particles with fewer neighbors. We choose $c = 40$, which corresponds to the average number of

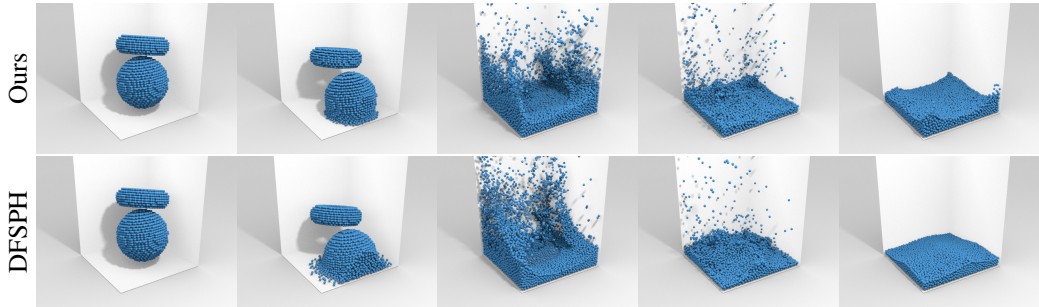

Figure 3: Comparison to ground-truth physics simulation. Two fluid bodies collide. Top: simulation by our trained network. Bottom: simulation of the same scenario by DFSPH (Bender & Koschier, 2015), a high-fidelity solver that was executed with small timesteps (down to 0.001s). Despite using a much larger timestep (0.02s), our convolutional network produces results of comparable visual fidelity. Note that our particles are falling slightly more slowly due to differences in the integration of positions and the much larger timestep. See the supplementary video.

neighbors across our experiments. Particles with few neighbors are close to the surface or interact with the scene boundary. Both cases are important for fluid simulation because particles near the surface define the liquid-air interface, which is particularly salient, and particles near the scene boundary require collision handling. The parameter $\gamma = 0.5$ makes our loss function more sensitive to small particle motions, which is important for increasing the accuracy and visual fidelity for small fluid flows. During training we predict particle positions for two future timesteps, namely $n+1$ and $n+2$. The combined loss $\mathcal{L}$ is

$$\mathcal{L} = \mathcal{L}^{n+1} + \mathcal{L}^{n+2}. \tag{17}$$

We found that optimizing a loss defined over two frames improves the overall quality of the simulation. (Optimization for three frames did not result in further improvements.) We optimize $\mathcal{L}$ over 50,000 iterations with Adam (Kingma & Ba, 2015) and a learning rate decay with multiple steps, starting with a learning rate of $0.001$ and stopping with $1.56 \cdot 10^{-5}$.

### 5.3 DATASETS

We have trained our network on multiple datasets. For quantitative comparisons with prior work we trained our network on the dam break data from Li et al. (2019). The scene simulates the behavior of a randomly placed fluid block in a static box. We generate 2000 scenes for training and 300 for testing. The data was generated with FleX, which is a position-based simulator that targets real-time applications (Macklin et al., 2014).

We also trained our network on more challenging data generated with DFSPH (Bender & Koschier, 2015), which prioritizes simulation fidelity over runtime. DFSPH can generate accurate fluid flows with very low volume compression: a desired property. We generate ground-truth data by randomly placing multiple bodies of fluid in 10 different box-like scenes and simulating them for 16 seconds each with an adaptive timestep of up to $1\,\mathrm{kHz}$. The time resolution of the generated data is $50\,\mathrm{Hz}$. We show a qualitative comparison of our method to the ground truth in Figure 3. We generate 200 scenes for training and 20 scenes for the test set. To train networks that can deal with multiple materials, we additionally generate 200 scenes with fluids of varying viscosity. For estimating material properties, we generate 7 test scenes that only differ in the viscosity parameter.

## 6 EVALUATION

**Baselines.** We compare our method to DPI-Nets (Li et al., 2019), which were previously shown to significantly outperform prior formulations such as hierarchical relation networks (Mrowca et al., 2018). Figure 4 provides a qualitative comparison. Quantitative results are reported in Table 1. To analyze the accuracy of the forward step for each method, we compute the average error of the particle positions with respect to the ground truth. We use every $5^{\text{th}}$ frame for initialization and compute the deviation from the ground truth for two subsequent frames, denoted by $n+1$ and $n+2$.

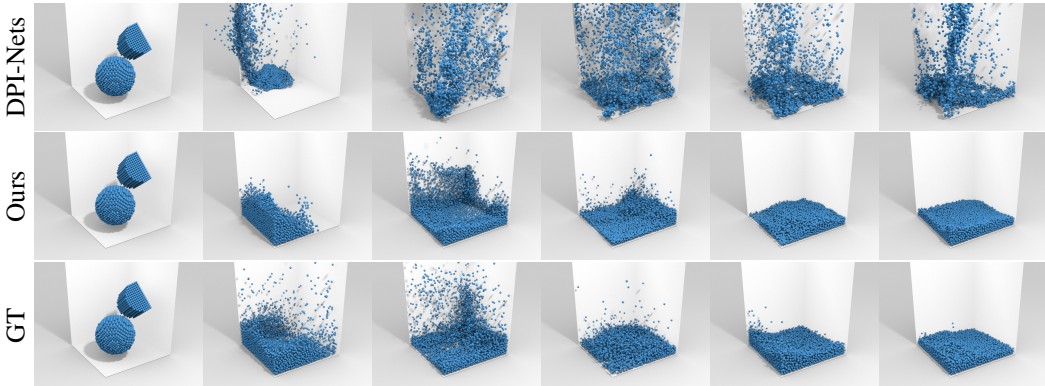

Figure 4: Qualitative comparison with DPI-Nets on a test sequence from our dataset. Two fluid bodies collide inside a container. DPI-Nets works well on data with little variance but has problems with more complex scenes and high particle velocities. The DPI-Nets simulation becomes unstable immediately after the fluid hits the box. The fluid behavior predicted by our network matches the ground truth more closely and remains stable for the whole sequence. The two networks have been trained on the same data. Test sequences are distinct from training sequences. See the supplementary video.

In addition, we report the average distance from the ground-truth particles to the closest particle in the prediction for the whole sequence, to measure long-term similarity. We compute the distance for frame $n$ as

$$d^n = \frac{1}{N} \sum_{i=1}^{N} \min_{\mathbf{x}^n \in X^n} \|\hat{\mathbf{x}}_i^n - \mathbf{x}^n\|_2 \,, \tag{18}$$

where $X^n$ is the set of predicted particle positions for frame $n$, $\hat{\mathbf{x}}_i^n$ is the ground-truth position for particle $i$, and $N$ is the number of particles.

We also compare our continuous convolution formulation to continuous convolution representations used in SPNets (Schenck & Fox, 2018), PCNN (Wang et al., 2018), KPConv (Thomas et al., 2019), and SplineCNN (Fey et al., 2018). To this end, we plug the respective convolution operators into our network architecture as shown in Figure 2. This facilitates controlled comparisons in which the overall architecture and simulation setup are fixed and only the convolution operators are varied. As shown in Table 1, our method outperforms all baselines with respect to both accuracy and inference time.

We have trained and tested all methods on the dam break dataset from Li et al. (2019) as well as our data (generated with a high-fidelity simulator (Bender & Koschier, 2015)). To facilitate the training on our data for all methods, we generated a simplified version with a constant number of particles (6,000) and a single box environment (as shown in Figure 4). We train DPI-Nets for 5 epochs and all other networks for 50,000 iterations, which corresponds to 2.7 epochs for the dam break data and 5 epochs for our data. Training takes about a day for our method with our convolutions on an NVIDIA RTX 2080Ti. Training with PCNN convolutions (Wang et al., 2018) takes about 2 days on 4 GPUs. Note that our reimplementation of PCNN convolutions uses Tensorflow's built-in functions, which consume a lot of memory and necessitate multi-GPU training. The DPI-Nets model trains in about one day. Training with SplineCNN Convs takes 3 to 4 days on a single GPU. We got the best results for this method with spherical kernel coordinates and closed splines. For KPConvs we used a Quadro RTX 6000 with 24 GB of RAM due to the higher memory requirements. Training took about 1 day with 15 kernel points. For SPNets convolutions, we estimated a training time of more than 29 days with $3 \times 3 \times 3$ filters by extrapolating from timing of a smaller number of iterations. We thus only report inference time for this method. Note that the convolutions of SPNets were designed to implement the position-based fluids algorithm, while we use them here in a more general network architecture with a much larger number of channels, which explains the very long runtime.

**Ablation study.** We perform an ablation study to evaluate our decisions in the design of the continuous convolution operator and the network. We study the importance of the interpolation, the window

| | Method | Average pos error (mm) | | Average distance to closest point $d^n$ (mm) | Frame inference time (ms) |
|---|---|---|---|---|---|
| | | $n+1$ | $n+2$ | | |
| DPI DamBreak | DPI-Nets | 12.73 | 25.38 | 22.07 | 202.56 |
| | SPNets Convs | – | – | – | 1058.46 |
| | PCNN Convs | 0.72 | 1.67 | 19.79 | 187.34 |
| | SplineCNN Convs | 0.71 | 1.65 | 170.20 | 67.67 |
| | KPConv | 2.49 | 7.05 | unstable | 47.96 |
| | Ours | **0.62** | **1.49** | **16.98** | **12.01** |
| Our data (6K particles) | DPI-Nets | 26.19 | 51.77 | unstable | 305.55 |
| | SPNets Convs | – | – | – | 784.35 |
| | PCNN Convs | 0.67 | 1.87 | 32.51 | 319.17 |
| | SplineCNN Convs | 0.68 | 1.93 | unstable | 281.92 |
| | KPConv | 1.65 | 4.54 | unstable | 57.89 |
| | Ours | **0.56** | **1.51** | **29.50** | **16.47** |

Table 1: Accuracy and runtime analysis. We compare the average error between the ground-truth particle positions and two predicted future frames on the test set. Additionally, we report the average distance from the ground truth to the prediction over the whole sequence. In this test mode some methods become unstable after a few frames. In the last column we report the average inference time per frame.

function, the loss design, and the architecture choice. The results are reported in Table 2. This table reports error measures (averaged over the test sequences) that were used in Table 1, and also reports the errors for two predicted frames initialized with the frames at the end of each sequence to measure the errors for small flow velocities. Large errors for small velocities can yield perceptually salient artifacts: rather than being still, fluid particles jitter or churn.

| Method | Average error (mm) | | Seq. end error (mm) | | Average distance to closest point $d^n$ (mm) |
|---|---|---|---|---|---|
| | $n+1$ | $n+2$ | $n+1$ | $n+2$ | |
| Ours | **0.67** | **1.87** | **0.25** | **0.74** | **30.63** |
| Ours w/o interpolation | 0.79 | 2.24 | 0.30 | 0.89 | 32.39 |
| Ours w/o window | 0.77 | 2.21 | 0.30 | 0.89 | 31.77 |
| Ours w/ naïve loss | 0.69 | **1.86** | 0.27 | 0.77 | **30.35** |
| Ours w/o FC | 0.75 | 2.17 | 0.27 | 0.80 | 32.49 |

Table 2: Ablation study. We compare the average error between the ground-truth particle positions and two predicted future frames on the test set, evaluated over whole test sequences (left) and just on the final frames of each test sequence (middle; this focuses on frames with small motion). The rightmost column shows the average distance from the ground truth to the predicted point set over whole test sequences. *Ours w/o interpolation* uses nearest-neighbor instead of trilinear interpolation for the convolution filters. *Ours w/o window* uses $a(\mathbf{x}_i, \mathbf{x}) = 1$ in Equation 7. *Ours w/ naïve loss* uses Euclidean distance as the loss, i.e. we set $\gamma = 1$ and $\phi_i = 1$ in Equation 16. *Ours w/o FC* uses only convolutions and includes the central particle in the convolution (rather than separately processing its features via an FC layer).

**Generalization.** In Figure 5 we show that our network generalizes well to scenes with drastically different geometry than seen during training. (The training set uses only box-like containers. See the appendix for visualization.) These scenarios demonstrate that we can emit particles during simulation, which can be costly for methods that build and maintain explicit graph structures. We compare generalization performance quantitatively to DFPSH on a complex scene in Figure 6. Figure 7 demonstrates generalization along a different dimension. Here we show that we can set the viscosity of the fluid at test time to a value not seen during training. The fluid shape used in this example was also not seen during training.

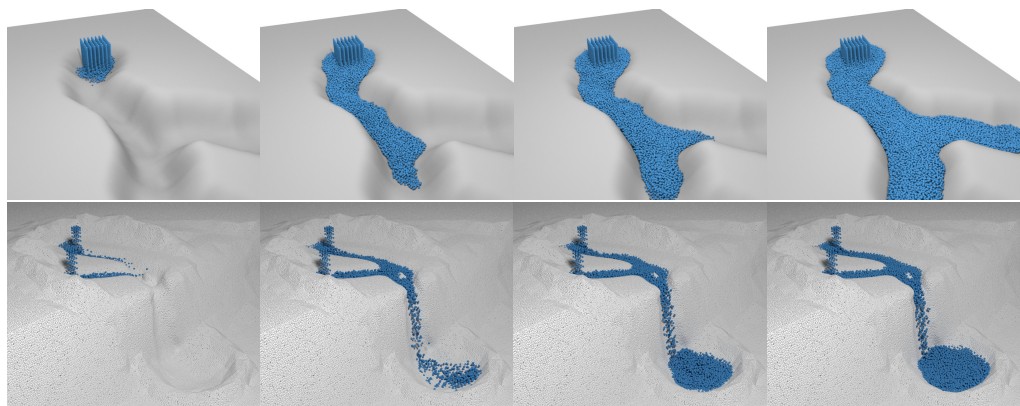

Figure 5: Generalization to environments with drastically different geometry than seen during training. Top: we use an emitter to fill up a virtual river with fluid particles, demonstrating generalization with respect to scene geometry and the number of particles. Bottom: a waterfall scene showing the fluid particles and the particle representation of the environment. See the supplementary video.

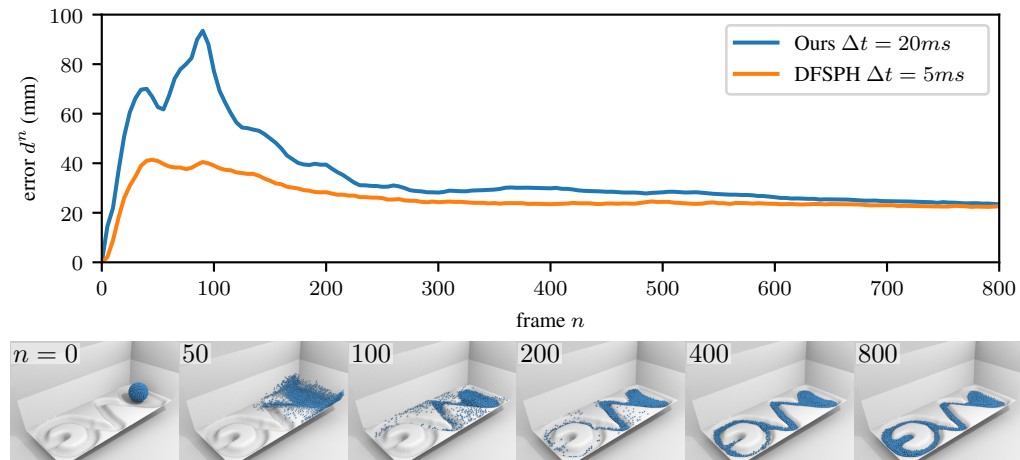

Figure 6: Average distance from the ground-truth particles to the predicted particles on a complex scene. Top: error over time for our trained network and DFSPH. We use a timestep of $\Delta t = 5\,\text{ms}$ for DFSPH and $\Delta t = 20\,\text{ms}$ for our method, which also corresponds to the frame sampling rate. The ground truth was generated with DFSPH and a timestep of 1ms. Bottom: simulation produced by our network. Large errors are concentrated in the beginning of the sequence when the fluid initially collides with the environment and the fluid behavior is most chaotic. During this phase the error is higher for our method than DFSPH. After 200 frames the error levels become similar.

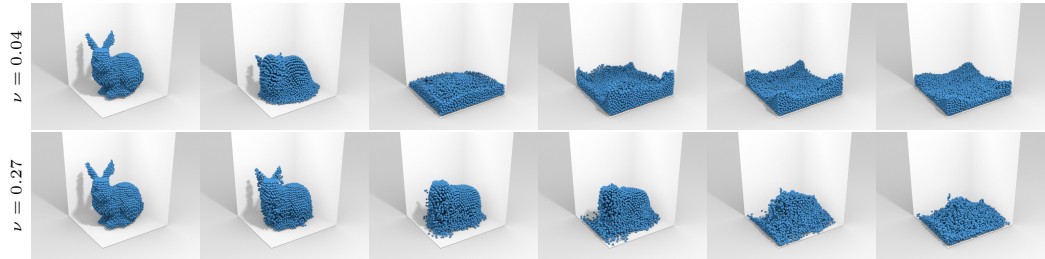

Figure 7: We can control the viscosity of the simulated fluid at test time by changing the input parameter $\nu$ in the input feature vector.

**Material estimation.** In this experiment we apply our network to an inverse problem: material estimation from observation. We use our network to estimate the viscosity parameter of a fluid from the particle movement. The training data for this experiment contains 200 sequences with random viscosity parameters between $0.01$ and $0.3$. For testing we generate 5 new sequences with 100 frames and random viscosity within the same range as used during training and use an initial fluid shape not present in the training data. To test generalization we generate two sequences with viscosity values outside the training range, namely $0.35$ and $0.4$. To estimate the viscosity we backpropagate through the trained network and optimize $\nu$ with gradient descent. Table 3 reports the results, which indicate that our network can be used to estimate material properties from observed data.

| GT viscosity | 0.044 | 0.127 | 0.174 | 0.233 | 0.269 | 0.350 | 0.400 | *Mean* |
|---|---|---|---|---|---|---|---|---|
| Avg. estimated viscosity | 0.027 | 0.150 | 0.202 | 0.255 | 0.277 | 0.322 | 0.336 | |
| Avg. relative error (%) | 38.377 | 18.542 | 15.604 | 9.568 | 3.235 | 7.954 | 16.016 | 15.614 |

Table 3: Application to an inverse problem: material estimation from observed fluid motion. To estimate the viscosity we backpropagate through our network and optimize $\nu$ with gradient descent. For each scene, we run the procedure 10 times, each time with random initialization, and report the average. Viscosity values $0.350$ and $0.400$ are outside the range that was used during training and are used to test generalization.

## 7 CONCLUSION

We have developed continuous convolutional networks for Lagrangian fluid simulation. We have introduced a simple formulation for continuous convolutions and demonstrated its accuracy and speed. Our model captures a wide range of complex material behavior, offers long-term stability, and generalizes to new situations such as varying particle counts, domain geometries, and material properties. There are numerous directions for future work, such as extending the framework to incorporate rigid and deformable solids. We will release the code to facilitate such development. Our continuous convolution implementation will be made available as part of Open3D (Zhou et al., 2018).

**Acknowledgements.** We thank Jan Bender for his support with the SPlisHSPlasH framework.

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

## A  APPENDIX

### A.1  IMPLEMENTATION DETAILS

To accelerate the computation of Equation 7 we use existing general matrix multiplication primitives. Similar to standard convolutions in deep learning frameworks we build a matrix with patches that we then multiply with the filter matrix. We can account for the linear interpolation by applying the interpolation to the patch matrix instead of the filters. For 3D point clouds a single point contributes to up to 8 voxels in a patch.

Another crucial part of our implementation is the nearest neighbor search. Since the positions of the fluid particles update with each timestep, we have to rebuild the neighborhood information for every frame. We implement the neighborhood search with spatial hashing (we use the hash function proposed in Teschner et al. (2003)). We explicitly store all particle neighbors in a compact list, which allows us to reuse the information for multiple convolutions operating on the same point sets. Table 4 compares the average frame runtimes of our method with the baselines.

### A.2  DATASET GENERATION

To enable generalization to new environments we use 10 different containers (see Figure 8) in our data generation process. During scene generation we randomly sample an environment and place up to 3 fluid bodies with different shapes and sizes (see Figure 9) and random initial velocities in the scene. We simulate each generated scene with DFSPH using the SPlisHSPlasH framework [1] for

---

[1] https://github.com/InteractiveComputerGraphics/SPlisHSPlasH

| | Method | Frame inference time (ms) | Frame NNS time (ms) | NNS Method |
|---|---|---|---|---|
| DPI DamBreak | DPI-Nets | 202.56 | 103.26 | KD-Tree (SciPy) |
| | SPNets Convs | 1058.46 | 5.24 | Spatial hashing on GPU |
| | PCNN Convs | 187.34 | 2.42 | *Spatial hashing on GPU |
| | SplineCNN Convs | 67.67 | 41.92 | Brute-force on GPU |
| | KPConv | 47.96 | 28.12 | KD-Tree (nanoflann) |
| | Ours | **12.01** | **2.14** | *Spatial hashing on GPU |
| Our data (6k particles) | DPI-Nets | 305.55 | 171.22 | KD-Tree (SciPy) |
| | SPNets Convs | 784.35 | 10.19 | Spatial hashing on GPU |
| | PCNN Convs | 319.17 | 2.78 | *Spatial hashing on GPU |
| | SplineCNN Convs | 281.92 | 245.52 | Brute-force on GPU |
| | KPConv | 57.89 | 34.07 | KD-Tree (nanoflann) |
| | Ours | **16.47** | **2.38** | *Spatial hashing on GPU |

Table 4: Runtime analysis. We compare the average per frame inference time and the time used for the nearest neighbor search (NNS). Our convolution achieves the shortest inference times in comparison even if NNS times would be excluded. Irrespective of that, the methods used for finding neighbors can have a significant contribution to the total runtime. Since fluid particles are moving, acceleration structures for the neighbor search have to be rebuilt each frame. This affects the KD-Tree methods as well as the methods using spatial hashing. Note that we use the same NNS for PCNN and Ours (denoted with *). For all methods except for SPNets the inference time is shorter on the smaller DPI DamBreak data (3456 particles compared to the 6000 particles of our data). We attribute this to a higher number of neighbors on the DPI DamBreak, which is about 49 on average compared to the average 40 neighbors on our datasets. All runtimes were measured on a system with an Intel Core i9-7960 and an NVIDIA RTX 2080Ti.

16 seconds to ensure that each scene contains frames with small particle velocities. To create the particle representation of the box-like containers, we do Poisson-Disc sampling on the mesh surface with the tools provided by DFPSH and add surface normals to each particle.

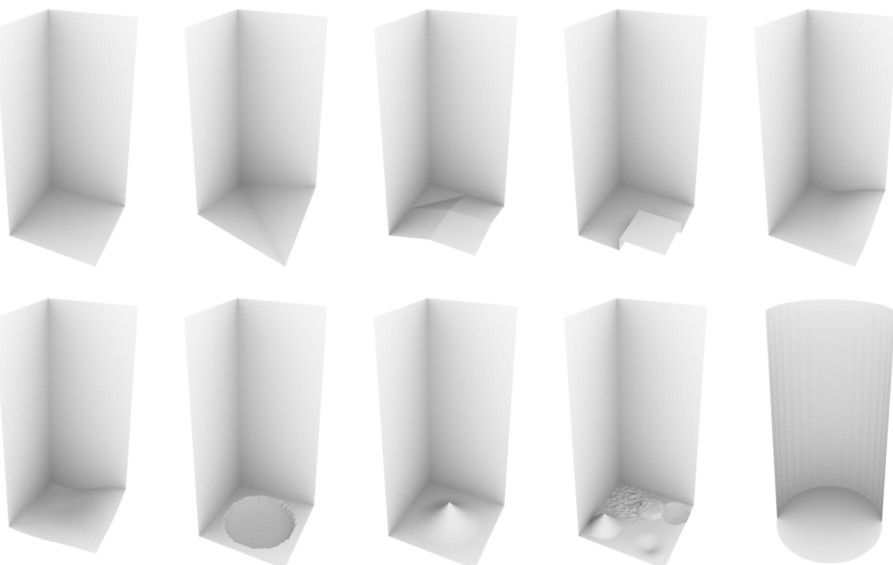

Figure 8: We sample from 10 different box-like containers during data generation. For the simplified version of our dataset used in the quantitative comparison with the baselines we only use the first container (leftmost container in the first row).

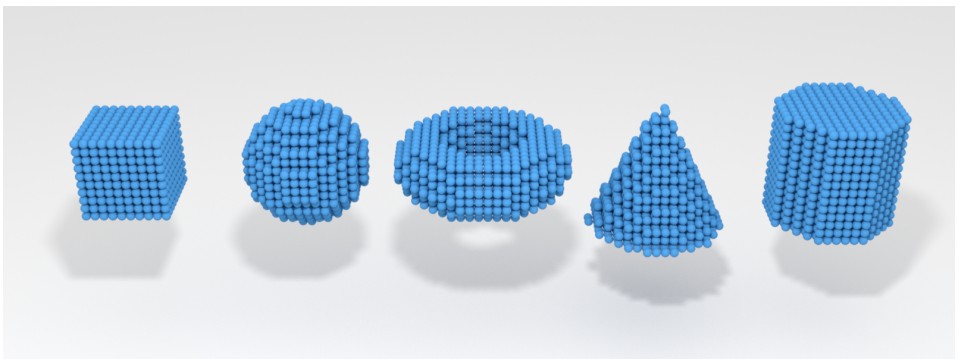

Figure 9: We randomly place fluid bodies of different initial shapes in the scene during data generation. We sample from 5 different shapes and vary the size, the orientation and the initial particle velocity. All particles from the same fluid body start with the same initial velocity. The image shows the particles generated from each shape for a specific size and orientation.

| Method | Average error (mm) | | Seq. end error (mm) | | Average distance to |
| --- | --- | --- | --- | --- | --- |
| | $n+1$ | $n+2$ | $n+1$ | $n+2$ | closest point $d^n$ (mm) |
| Ours | **0.67** | **1.87** | **0.25** | **0.74** | **30.63** |
| Ours triangular window | 0.69 | 1.94 | 0.27 | 0.79 | **30.28** |
| Ours w/o window | 0.77 | 2.21 | 0.30 | 0.89 | 31.77 |

Table 5: Comparison of different window functions. *Ours* uses a window function similar to the poly6 kernel used in Müller et al. (2003). *Ours triangular window* uses a triangular window function. *Ours w/o window* does not use a window function. This case can also be interpreted as a rectangular window since we only consider points within a radius $R$.

## A.3 TRAINING DETAILS

We use the Tensorflow framework for implementing the training procedure. We use Adam as optimizer and train with a batch size of 16 and an initial learning rate of 0.001. We half the learning rate at steps 20000, 25000, ..., 45000. For the convolutions we use the random uniform initializer with range [-0.05, 0.05]. All other weights are initialized with the respective default initializers of Tensorflow version 1.12. The output of the network is scaled with $\frac{1}{128}$ to roughly adjust the output range to the ground truth position correction of the training data.

The unit of length of the training data is meter. The particle radius used in DFSPH is $h = 0.025\,\mathrm{m}$. For our convolutions, we use spherical filters with an empirically determined radius of $R = 4.5h$.

## A.4 WINDOW FUNCTION

We compare 3 different choices for the window function $a$ from Equation 8 in Table 5. The results show that enforcing a continuous output with a window function yields better results. Further, using a window function similar to kernels used in SPH codes gives better results than a simple triangular window. Learning the window function is therefore a possible direction to extend our framework to further improve the accuracy.

## A.5 COORDINATE MAPPING FUNCTION

We use the ball to cube mapping described in (Griepentrog et al., 2008) to map a position within a spherical region to the filter values stored in a regular grid. We give here the functions used for the 3-D case as used in our implementation. For more details see (Griepentrog et al., 2008).

The function $\Lambda$ is a composition of the functions $\Lambda_{\text{ball}\to\text{cyl}}$ and $\Lambda_{\text{cyl}\to\text{cube}}$, which map a sphere to a cylinder and a cylinder to a cube respectively. We define $\Lambda_{\text{ball}\to\text{cyl}}$ for vectors $\mathbf{r} = (x, y, z)$ as

$$\Lambda_{\text{ball}\to\text{cyl}}(\mathbf{r}) = \begin{cases} (0, 0, 0) & \text{if } \|\mathbf{r}\|_2 = 0 \\ \left(x\frac{\|\mathbf{r}\|_2}{\|(x,y)\|_2}, y\frac{\|\mathbf{r}\|_2}{\|(x,y)\|_2}, \frac{3}{2}z\right) & \text{if } \frac{5}{4}z^2 \leq x^2 + y^2 \\ \left(x\sqrt{\frac{3\|\mathbf{r}\|_2}{\|\mathbf{r}\|_2 + |z|}}, y\sqrt{\frac{3\|\mathbf{r}\|_2}{\|\mathbf{r}\|_2 + |z|}}, \text{sign}(z)\|\mathbf{r}\|_2\right) & \text{else.} \end{cases} \tag{19}$$

The cylinder to cube mapping is defined as

$$\Lambda_{\text{cyl}\to\text{cube}}(\mathbf{r}) = \begin{cases} (0, 0, z) & \text{if } x = 0, y = 0 \\ \left(\text{sign}(x)\|(x,y)\|_2, \frac{4}{\pi}\text{sign}(x)\|(x,y)\|_2 \arctan\frac{y}{x}, z\right) & \text{if } |y| \leq |x| \\ \left(\frac{4}{\pi}\text{sign}(y)\|(x,y)\|_2 \arctan\frac{x}{y}, \text{sign}(y)\|(x,y)\|_2, z\right) & \text{else.} \end{cases} \tag{20}$$

We assume that vectors $\mathbf{r}$ are normalized with the search radius $R$ such that $\|\mathbf{r}\|_2 \leq 1$. This yields the following $\Lambda$, which maps from a unit ball to the normalized coordinates of a cube:

$$\Lambda(\mathbf{r}) = \frac{1}{2}\Lambda_{\text{cyl}\to\text{cube}}(\Lambda_{\text{ball}\to\text{cyl}}(\mathbf{r})) + (0.5, 0.5, 0.5). \tag{21}$$

