# OpenReview forum: "Lagrangian Fluid Simulation with Continuous Convolutions"
_ICLR.cc/2020/Conference — Accept (Poster)_

### Official Review · AnonReviewer4 · 2019-10-23
**Official Blind Review #4**

**Rating:** 6

**Review:**

This paper applies 3D convolutions to the problem of Lagrangian fluid simulation. The primary difficulty in this is that, unlike for Eulerian fluid simulation, which represents the fluid as a grid and adapts nicely to 3D convolutions, in Lagrangian simulations the fluid is represented as an unordered set of particles. It is not straight-forward to apply 3D convolutions on such a data structure, however this paper proposes a method to apply the same regular-grid kernels used in grid-based convolutions to the particle structure. To do this, several points around the kernel are evaluated by first using trilinear interpolation between the particles to get feature values at those points and then convolving those values with the kernel weights. This results in a new particle in the next layer up with those features. In the paper, this method is used to train the weights of the network to reproduce fluid dynamics generated by a simulator. The results show that the proposed method was able to model fluid dynamics over 2 timesteps more accurately than other methods and can do so quickly.

While I have some reservations about this paper (detailed below), on the whole I think it is a quality contribution and should be accepted. This paper contributes a novel method for performing 3D convolutions on unordered particle sets, and it shows that the learned fluid dynamics generalize to novel situations. One major hurdle to applying modern convolutional learning techniques to Lagrangian methods is the mismatch between the layout of the data (unordered particles) and the layout of the kernels (regular grid). This paper presents a novel way of bridging that divide, and it shows that the proposed method actually works by applying it to the problem of fluid dynamics and successfully learning it. However, one major concern I had was that it seems all of the training data was generated in box-like environments, which could easily lead to overfitting. This was alleviated by the results showing that although the network was trained only in boxes, it generalized to environments with channels and waterfalls (as seen in the video). This is a powerful result and shows that this method really did learn fluid dynamics and not just a shortcut that only works in boxes.

I do think this paper can be improved in a few aspects however. The biggest issue is that the quantitative analysis of the core functionality (reproducing fluid physics) is lacking. The paper only reports results for error after at most 2 timesteps, which is not nearly long enough to determine if the output is accurate. Furthermore, the results are only reported for the box scenes, not the generalization scenes mentioned above. Qualitatively, from the videos, it is clear that the output does at least somewhat model fluid dynamics, but it would be much better to have hard numbers to back that up. I suspect the authors discovered that Lagrangian systems are sufficiently chaotic that after only a few timesteps the particle positions have diverged significantly. This is not a bug but a feature of such systems. In Lagrangian fluids, the particles are but an approximation of the fluid, and unlike Eulerian systems*, multiple different sets of particles can approximate the same fluid. This makes particle position only useful as a measure of error if the trained model can perfectly reproduce the fluid dynamics. But of course it can't (trained networks aren't ever perfect in practice), and so small errors quickly compound into large particle position disparities. So even though the trained network models the fluid well overall, the particles end up in completely different locations. Instead a better error metric would be something like measuring the difference between the surface of the fluids, or the velocities or densities at various locations. These are agnostic to the particular particle positions, but still measure how well two different sets of particles represent the same fluid. Using a metric like this, it would be nice to see error graphs over time for both the box and generalization scenes.

A couple other smaller points. The chaotic divergence behavior of DPI-Nets seems inconsistent with that paper. Is this possibly a bug in the way it was implemented here? Additionally, the paper states that the convolutions of SPNets were "specifically designed to implement the position-based fluids algorithm" but that it was used in the paper with a much larger number of channels. If it was designed only to work for that one algorithm, how were the number of channels increased? That is unclear. Also, the average error for SPNets is not shown in Table 1 and it is not stated why.

*Assuming same grid shape, size, and position.

**Experience Assessment:**

I have published in this field for several years.

**Review Assessment: Checking Correctness Of Derivations And Theory:**

I assessed the sensibility of the derivations and theory.

**Review Assessment: Checking Correctness Of Experiments:**

I carefully checked the experiments.

**Review Assessment: Thoroughness In Paper Reading:**

I read the paper thoroughly.

---

> ### Author Response · Authors · 2019-11-13
> **Response to Reviewer 4**
>
> We thank R4 for the comments and suggestions.
>
> Q: “However, one major concern I had was that it seems all of the training data was generated in box-like environments, which could easily lead to overfitting. This was alleviated by the results showing that although the network was trained only in boxes, it generalized to environments with channels and waterfalls (as seen in the video). This is a powerful result and shows that this method really did learn fluid dynamics and not just a shortcut that only works in boxes.”
>
> A: Generalization is a major challenge in learning physics. We thank R4 for this comment.
>
>
> Q: “The biggest issue is that the quantitative analysis of the core functionality (reproducing fluid physics) is lacking. The paper only reports results for error after at most 2 timesteps, which is not nearly long enough to determine if the output is accurate. Furthermore, the results are only reported for the box scenes, not the generalization scenes mentioned above. Qualitatively, from the videos, it is clear that the output does at least somewhat model fluid dynamics, but it would be much better to have hard numbers to back that up.”
>
> A: Yes, the evaluation and comparison of chaotic flows is a challenging topic by itself. To provide more information about the performance of our method, we added a new metric to capture the similarity of the fluids over the whole sequence. We measure the distance from the ground truth point cloud to the closest points in the predicted point cloud and added the new metric to table 1 and 2. Our approach compares favourably to the baselines with the new metric.
> We also added another generalization scene for which we plot the error over time. To better assess the numbers we also evaluate and add DFSPH with a larger time step as a reference. The experiment shows that our method reaches a similar accuracy as DFSPH for the majority of frames. We present the results in Figure 6 in the updated paper.
>
>
> Q: “The chaotic divergence behavior of DPI-Nets seems inconsistent with that paper. Is this possibly a bug in the way it was implemented here?”
>
> A: We use the publicly available code for training DPI-Nets on both datasets. We uploaded a video with a qualitative comparison on a sequence of the dam break dataset, which shows that DPI-Nets is stable despite being less accurate. However, on our data set with more violent motions the DPI-Net is less accurate and unstable. The comparison is shown at the bottom of https://sites.google.com/view/lfswcc .
>
>
> Q: “Additionally, the paper states that the convolutions of SPNets were "specifically designed to implement the position-based fluids algorithm" but that it was used in the paper with a much larger number of channels. If it was designed only to work for that one algorithm, how were the number of channels increased? That is unclear. Also, the average error for SPNets is not shown in Table 1 and it is not stated why.”
>
> A: We use the convolutions from SPNets with our network architecture to compare the performance to our continuous convolution implementation. We made this more clear in the updated paper. While the SPNets convolutions were designed with the PBF algorithm in mind, the implementation is quite general and allows to change the number of channels. However, we measure very long runtimes using the convolutions in our more general training scenario. We do not state the average error because we estimate a training time of at least 29 days. We state this in the updated paper and added more comparisons with other state-of-the-art convolutions instead.

---

### Official Review · AnonReviewer3 · 2019-10-23
**Official Blind Review #3**

**Rating:** 8

**Review:**

[Summary]

This paper proposes to learn fluid dynamics by combining the position-based fluids (PBF) framework and continuous convolution. They use dynamics particles to represent the fluids, and static particles to describe the scene boundaries, and employ continuous convolution to learn the interactions between the particles of different kinds. They have demonstrated the effectiveness of the proposed method by comparing it with several state-of-the-art learning-based and physics-based fluid simulators. Their method outperforms the baselines in terms of both accuracy and efficiency. They have also shown that the model can extrapolate to terrains that are more complex than those used in training, and are useful in estimating physical properties like the viscosity of the fluids.


[Major comments]

For now, I slightly lean towards acceptance, as I like the idea of combining PBF and continuous convolution for fluid simulation, and the method seems to have a much better performance than the baselines. The experiments have also convincingly demonstrated the method's generalization ability to terrains of various geometry and fluids of different viscosity. However, I would still like the authors to address my following questions.

My primary concern about the proposed method is the scope of its applicability. One of the benefits of using learning-based physics engines is that they directly learn from observations while making very few assumptions towards the underlying dynamics, which gives them the potential to handle complex real-world scenarios. The model in this paper, however, heavily relies on the PBF framework that may limit its ability to simulate objects like rigid bodies and other deformable materials. I would be curious to know the authors' views on how to extend their model to environments with not just fluids, but also other objects of various material properties.


[More detailed questions]

Will the method run faster than DFSPH, given that the timestep is much larger than the timestep used by DFSPH, 0.02 ms vs. 0.001 ms? Will the learning-based physics engine have the potential to outperform the physics-based physics engine in terms of efficiency?

For estimating the viscosity of the fluids, how well does the gradient descent on the learned model perform comparing with black-box optimization, e.g., Bayesian Optimization using the ground truth simulator?

In the SPNet paper, they have also tried to solve the inverse problem of estimating the viscosity of the fluids. It would be great to include a comparison to see if the proposed method can outperform SPNet in terms of efficiency and accuracy.

Equation 8 smooth out the effect between particles of different distances. How sensitive is the final performance of the model to the specific smoothing formulation? Is it possible to learn a reweighting function instead of hardcoding?

In figure 3, the model's rollout is a bit slower than the ground truth. The authors explained the phenomenon using the "differences in the integration of positions and the much larger timestep." I do not quite get the point. Could you elaborate more on this? Also, it might be better to include labels for the two columns in figure 3 to make it more clear.

In the experiment section, the authors claimed that SPNets take "more than 29 days" to train. Correct me if I am wrong, but from my understanding, SPNets directly write Position-Based Fluids (PBF) in a differentiable way, where they can extract gradients. Except for the tunable parameters like viscosity, cohesion, etc., I'm not sure if there are any learnable parameters in their model. Could the authors elaborate on what they mean by "the training time" of SPNets?

From the videos, DPI-Nets does not seem to have a good enough performance in the selected environments. I can see why their model performs not as good since they did not use as much of a structure in the model. But from the videos of DPI-Nets, it seems that they perform reasonably well in scenes like dam break or shake a box of fluids. Would you please provide more details on why they are not as good in the scenes in this paper?

The data was generated using viscosity varying between 0.01 and 0.3. How well can the model do extrapolate generalization? It would be great to show some error plots indicating its extrapolate performance.

Why there are no average error numbers for SPNets?


**Experience Assessment:**

I have published one or two papers in this area.

**Review Assessment: Checking Correctness Of Derivations And Theory:**

I assessed the sensibility of the derivations and theory.

**Review Assessment: Checking Correctness Of Experiments:**

I carefully checked the experiments.

**Review Assessment: Thoroughness In Paper Reading:**

I read the paper thoroughly.

---

> ### Author Response · Authors · 2019-11-13
> **Response to Reviewer 3 (1/2)**
>
> We thank R3 for the comments and questions.
>
> [Major comments]
> Q: “My primary concern about the proposed method is the scope of its applicability. One of the benefits of using learning-based physics engines is that they directly learn from observations while making very few assumptions towards the underlying dynamics, which gives them the potential to handle complex real-world scenarios. The model in this paper, however, heavily relies on the PBF framework that may limit its ability to simulate objects like rigid bodies and other deformable materials. I would be curious to know the authors' views on how to extend their model to environments with not just fluids, but also other objects of various material properties.”
>
> A: The components that our approach has in common with the PBF framework are very general and apply to a vast range of simulations. Like PBF we work on positions and we extrapolate the particle positions based on the velocity and external forces for the next timestep.
> However, the remaining (interesting) parts of PBF like density constraints, vorticity confinement and collision handling, which actually implement the physical model and define how particles interact with each other are entirely learned from data in our approach.
> Therefore we think extensions such as the simulation of rigid bodies and deformable objects are possible with our framework. We already rely on the network to handle collisions with complex rigid environments, which indicates the possibility to simulate other materials. We see this as a future direction for applications of our method.
>
>
> [More detailed questions]
> Q: Will the method run faster than DFSPH, given that the timestep is much larger than the timestep used by DFSPH, 0.02 ms vs. 0.001 ms? Will the learning-based physics engine have the potential to outperform the physics-based physics engine in terms of efficiency?
>
> A: Yes, the method runs faster. For instance the scene shown in Figure 3 runs in real-time with our method while DFSPH needs about 9 minutes to simulate a sequence of 16 seconds. Our method is therefore more efficient with respect to the runtime. Following this direction, potential applications would be to approximate even more sophisticated SPH codes targeting CFD applications.
>
>
> Q: For estimating the viscosity of the fluids, how well does the gradient descent on the learned model perform comparing with black-box optimization, e.g., Bayesian Optimization using the ground truth simulator?
>
> A: We get a relative error of 4.5% with the hyperopt package. We run the optimizer for 21 iterations, which corresponds to the time budget used in our experiment. Since we only estimate a single scalar the problem works well with blackbox optimization. For more high dimensional problems, e.g. individual parameters for each particle, blackbox optimization becomes quickly infeasible.
>
>
> Q: In the SPNet paper, they have also tried to solve the inverse problem of estimating the viscosity of the fluids. It would be great to include a comparison to see if the proposed method can outperform SPNet in terms of efficiency and accuracy.
>
> A: Since SPNets implements PBF and we learn the behaviour of fluids with different viscosities from DFSPH the viscosity parameters are not compatible and unfortunately cannot be compared.
>
>
> Q: Equation 8 smoothes out the effect between particles of different distances. How sensitive is the final performance of the model to the specific smoothing formulation? Is it possible to learn a reweighting function instead of hardcoding?
>
> A: The type of the window function influences the performance. We cannot (yet) backpropagate to the window function but this is a reasonable extension. We added a comparison with a triangular window to the appendix.
>
>
> Q: “In figure 3, the model's rollout is a bit slower than the ground truth. The authors explained the phenomenon using the "differences in the integration of positions and the much larger timestep." I do not quite get the point. Could you elaborate more on this? Also, it might be better to include labels for the two columns in figure 3 to make it more clear.”
>
> A: Since DFSPH uses a much smaller time step it updates the particle velocities and positions more often resulting in slightly faster falling particles. Additionally, the time integration scheme is different. We use the midpoint method for computing the position, which is not used by DFPSH. Instead DFSPH corrects the density before updating the positions.
> We added labels to figure 3.

---

> > ### Author Response · Authors · 2019-11-13
> > **Response to Reviewer 3 (2/2)**
> >
> > Q: “In the experiment section, the authors claimed that SPNets take "more than 29 days" to train. Correct me if I am wrong, but from my understanding, SPNets directly write Position-Based Fluids (PBF) in a differentiable way, where they can extract gradients. Except for the tunable parameters like viscosity, cohesion, etc., I'm not sure if there are any learnable parameters in their model. Could the authors elaborate on what they mean by "the training time" of SPNets?”
> >
> > A: We use different implementations of continuous convolutions with our network architecture and compare them. This means we use the convolutions from SPNets in our architecture which uses a larger number of channels. We made this more clear in the revision. We measure very long runtimes for this convolution implementation in our more general network training scenario. We verified that the number of neighbors is comparable and that the GPU is fully utilized.
> >
> >
> > Q: “Why there are no average error numbers for SPNets?”
> >
> > A: Because of the very long training time we decided to use resources to add comparisons to other state-of-the-art continuous convolutions in the revised version.
> >
> >
> > Q: “From the videos, DPI-Nets does not seem to have a good enough performance in the selected environments. I can see why their model performs not as good since they did not use as much of a structure in the model. But from the videos of DPI-Nets, it seems that they perform reasonably well in scenes like dam break or shake a box of fluids. Would you please provide more details on why they are not as good in the scenes in this paper?”
> >
> > A: DPI-Nets performs worse than our method on the dam break sequence but it is stable. We uploaded a new video to the supplemental material at https://sites.google.com/view/lfswcc which shows a qualitative comparison on one of the dam break sequences. Please see the link to the supplementary material in the paper. On our data DPI-Nets is less accurate and unstable. We used the same code for training DPI-Nets on both datasets. Note that the video on the DPI-Nets homepage also shows some small artifacts for BoxBath #1, which indicates problems with stability similar to ones we observe in our tests.
> >
> >
> > Q: “The data was generated using viscosity varying between 0.01 and 0.3. How well can the model do extrapolate generalization? It would be great to show some error plots indicating its extrapolate performance.”
> >
> > A: We added 2 more sequences with viscosity parameters outside of the training range.

---

> > > ### Comment · AnonReviewer3 · 2019-11-13
> > > **Thank you for address my concerns.**
> > >
> > > I thank the authors for addressing my concerns, and the revisions strengthen the paper. I would like to raise my rating from 6 (Weak Accept) to 8 (Accept).

---

### Official Review · AnonReviewer1 · 2019-10-27
**Official Blind Review #1**

**Rating:** 6

**Review:**

This paper proposes a novel technique to perform fluid simulations. Specifically, they promote the idea of using spatial convolutions to model how particles interact with nearby particles. Compared to graph-based models, this approach has several advantages and yields a model that can be conveniently trained end-to-end. The authors also develop a specific type of continuous convolution that yield better and faster inference than the benchmark algorithms.

To main contribution in this paper is the idea of using spatial convolutions to model particle interactions. Even though the obtained results contain significant errors compared to ground truth, the paper indicates a promising strategy others may leverage on to develop even more accurate deep learning based simulators. Considering that they have also plausibly argued that their specific algorithm is already state of the art, I view this as a significant contribution. Having said this, the contribution is really to use a well-known technique (spatial convolutions) on a new problem (fluid simulations). My understanding is that ICLR primarily wants to promote general learning techniques and I am not convinced that this paper contains any significant contributions in this field.

The authors also develop a specific network architecture that they compare with other deep learning architectures for continuous convolutions. Unfortunately, the design contains a number of questionable choices and I suspect that the main reason that existing architectures for deep learning using continuous convolutions perform worse is that their hyper-parameters have been fine-tuned for a different task. As an example of a questionable choice, why do you “exclude the particle at which we evaluate the convolution” in your convolutions?

Minor remarks:
* You include a constant 1 in the input feature vectors. Assuming that the neurons in your network have weights and biases, this constant is completely redundant. Of course, this also means that you can include it without ruining your performance, but why would you?
* I found the explanation of Lambda in Figure 1 too short to be understandable.
* In (7), you seem to be using convolutions between functions that have not been pre-mirrored, and it would be better to then express (5) and (6) on the same form.

**Experience Assessment:**

I have read many papers in this area.

**Review Assessment: Checking Correctness Of Derivations And Theory:**

I assessed the sensibility of the derivations and theory.

**Review Assessment: Checking Correctness Of Experiments:**

I did not assess the experiments.

**Review Assessment: Thoroughness In Paper Reading:**

I read the paper at least twice and used my best judgement in assessing the paper.

---

> ### Author Response · Authors · 2019-11-13
> **Response to Reviewer 1**
>
> We thank R1 for the assessment and the comments for improving the paper.
>
> Q: “Having said this, the contribution is really to use a well-known technique (spatial convolutions) on a new problem (fluid simulations). My understanding is that ICLR primarily wants to promote general learning techniques and I am not convinced that this paper contains any significant contributions in this field.”
>
> A: While spatial convolutions are indeed well established, there is still no agreement on how to do spatial convolutions for unordered data like point clouds. In fact, the research community is quite active on this topic  (we list selected publications from 2018/2019 below [1-11]). We added more comparisons to other state-of-the-art convolutions to our evaluation in the revised version of the paper. Our network consistently yields an improved accuracy with a lower runtime.
> To our understanding ICLR accepts papers covering all aspects of deep learning. The list of topics explicitly mentions applications, and in our evaluation we compare with DPI-Nets, which was presented at ICLR 2019.
>
>
> Q: “The authors also develop a specific network architecture that they compare with other deep learning architectures for continuous convolutions. Unfortunately, the design contains a number of questionable choices and I suspect that the main reason that existing architectures for deep learning using continuous convolutions perform worse is that their hyper-parameters have been fine-tuned for a different task. As an example of a questionable choice, why do you “exclude the particle at which we evaluate the convolution” in your convolutions?”
>
> A: We added an experiment to the ablation study that explains this design choice. The choice is motivated by using a kernel with an even size and having dedicated weights for processing the features of the particle itself.
> To have a fair comparison with other continuous convolutions we use the convolutions in the same architecture. We made this more clear in the revision. For the parameters, we make sure that all methods have the same receptive field.
> For the newly added comparison with SplineCNN we evaluated Cartesian kernels and all options for spherical kernel parameterizations and selected the one that performed best.
> For the newly added comparison with KPConv we use 15 kernel points as suggested in their paper for the Scannet benchmark. Unfortunately, the official implementation has very high memory requirements and we could not fit settings with more kernel points on a GPU with 24GB of RAM.
> The PCNN convolution has been used for regression tasks on point clouds and performs well in our experiments.
>
>
> Minor remarks:
> Q: You include a constant 1 in the input feature vectors. Assuming that the neurons in your network have weights and biases, this constant is completely redundant. Of course, this also means that you can include it without ruining your performance, but why would you?
>
> A: We apply the bias after the convolutions, which means for the first convolution in the network that a point with a zero feature vector has no influence on the convolution result. For the following convolutions the network can learn a bias which replaces the constant 1 to identify points.
>
>
> Q: I found the explanation of Lambda in Figure 1 too short to be understandable.
>
> A: Thank you for pointing this out, we extended the explanation in the revision and added the detailed definition of the function to the appendix.
>
>
> Q: In (7), you seem to be using convolutions between functions that have not been pre-mirrored, and it would be better to then express (5) and (6) on the same form.
>
> A: In (7) we compute $x_i - x$ to get a relative position, which corresponds to $\tau$. We removed “pre-mirrored” from the text as we explicitly refer to convolutions in ConvNets.
>
>
> [1] Thomas et al., “KPConv: Flexible and Deformable Convolution for Point Clouds,” ICCV, 2019.
> [2] Liu et al., “Point-Voxel CNN for Efficient 3D Deep Learning,” NeurIPS, 2019.
> [3] Lei et al., “Octree guided CNN with spherical kernels for 3D point clouds,” CVPR, 2019.
> [4] Xu et al., “SpiderCNN: Deep Learning on Point Sets with Parameterized Convolutional Filters,” ECCV, 2018.
> [5] Wang et al., “Deep Parametric Continuous Convolutional Neural Networks,” CVPR, 2018.
> [6] Su et al., “SPLATNet: Sparse Lattice Networks for Point Cloud Processing,” CVPR, 2018.
> [7] Schenck and Fox, “SPNets: Differentiable Fluid Dynamics for Deep Neural Networks,” CoRL, 2018.
> [8] Li et al., “PointCNN: Convolution On X-Transformed Points,” NeurIPS, 2018.
> [9] Hermosilla et al., “Monte Carlo Convolution for Learning on Non-uniformly Sampled Point Clouds,” ACM Trans. Graph., vol. 37, no. 6, 2018.
> [10] Fey et al., “SplineCNN: fast geometric deep learning with continuous b-spline kernels,” CVPR, 2018.
> [11] Atzmon et al., “Point Convolutional Neural Networks by Extension Operators,” ACM Trans. Graph., vol. 37, no. 4, 2018.

---

### Author Response · Authors · 2019-11-13
**Changes in the revision**

Changes in the revision

We thank the reviewers for their help to improve the paper. In the following we list the changes to the draft:

* We extended the evaluation by adding KPConv convolutions and SplineCNN convolutions.
* We add a new metric to measure the distance between the GT and the prediction over the whole sequence.
* The ablation study now contains a version of our network without FC layers.
* We added the waterfall scene to Figure 5 as an example for the particle representation of the environment.
* We added a quantitative generalization experiment (Figure 6).
* We added 2 more test scenes to the viscosity estimation experiment to test generalization.
* We moved the runtime evaluation for the nearest neighbor search to the appendix and give more information.
* We discuss the choice of the window function in the appendix and report numbers for a triangular window.
* We added a figure showing the fluid shapes used for data generation in the appendix
* We give the definition of the function Lambda in the appendix

Minor changes:
* In equation 8 the window function is now normalized with respect to the radius.
* We improved the description of Figure 1
* Figure 2 now uses the same symbol for the viscosity as the main text.
* We added labels to Figure 3
* We updated and fixed missing information for 2 references.

---

### Decision · Program_Chairs · 2019-12-19

**Decision:**

Accept (Poster)

**Comment:**

The paper proposes an approach for N-D continuous convolution on unordered particle set and applies it to Lagrangian fluid simulation. All reviewers found the paper to be a novel and useful contribution towards the problem of N-D continuous convolution on unordered particles. I recommend acceptance.